# An Effective Data Sharing Scheme Based on Blockchain in Vehicular Social Networks

**Yanji Jiang** [1,2,*], **Xueli Shen** [2] **and Sifa Zheng** [1]

1    Suzhou Automotive Research Institute, Tsinghua University, Suzhou 125000, China; zsf@tsinghua.edu.cn
2    Software College, Liaoning Technology University, Huludao 125000, China; shenxueli@lntu.edu.cn
*    Correspondence: jyjvip@126.com

**Abstract:** Vehicular social networks (VSNs) are the vehicular ad hoc networks (VANETs) that integrate social networks. Compared with traditional VANETs, VSNs are more suitable to serve a group of vehicles with common interests. In VSNs, vehicles can upload the necessary data in the cloud service provider (CSP) and other vehicles can query the data they are interested in through CSP, which enables VSNs to provide more user-friendly services. However, due to the wireless network communication environment, the data sent by the vehicle can easily be monitored. Adversaries are able to violate the privacy of the vehicle based on the collected data, thereby threatening the security of the entire network. In addition, if a vehicle shares malicious or false data with other vehicles, it is easy to mislead drivers and even cause serious traffic accidents. This paper proposes an effective data sharing scheme based on blockchain in VSNs. By integrating an identity based signature mechanism and pseudonym generation mechanism, we first propose an anonymous authentication mechanism as the basis for establishing trust relationships before data transmission between entities in VSNs. Then, a data sharing scheme based on blockchain is described, in which the signature mechanism and the consensus mechanism guarantee the security and traceability of data. The result of the performance analysis and the simulation experiment indicate that VAB can achieve a favourable performance compared with existing schemes.

**Keywords:** vehicular social networks; authentication; data sharing; blockchain

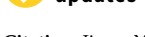



## 1. Introduction

Vehicular ad hoc networks (VANETs) are special mobile ad hoc networks (MANETs), which provide network and communication services for vehicles running on the road [1]. Due to the rapid movement of vehicles and wireless communication environment, VANETs have to confront the issues of dynamic changes of network topology, uneven density distribution of communication nodes, and noise jamming [2]. Recently, the United States and Europe propose independent standards to meet the communication requirements of VANETs, and offer a series of suggestions to solve the above problems [3,4]. Now, VANETs have been able to support a variety of services, such as traffic management, collision warning, and sharing data etc., which effectively promotes the development of VANETs in the world [5]. Vehicular Social Networks (VSNs) are thought as the integration between VANETs and social networks, which emphasizes the social attributes of VANETs [6]. A VSN is divided into several independent groups, in which each group has common interests or needs in a short time and the group members are close to each other on the road. For example, a group of people driving to the concert are looking forward to sharing data about the concert (i.e., traffic conditions near the stadium, guests of the concert, schedule, etc.). VSNs are required to be able to identify socially-similar vehicles and provide sharing information service.

### 1.1. Problem Definition

Due to wireless network environment, VSN has to face a variety of cyber threats and challenges [7]. In the process of data sharing, external attackers can easily collect the transmitted data stream. If there is no effective mechanism to protect the security of the data, the external attacker can not only obtain the information contained in the data stream, but also violate the privacy information of data sender, such as driving track, hobbies, etc. In addition, if an illegal vehicle joins a group, the vehicle can also send fake messages (traffic flow of the target road) to confuse other vehicles, which is easy to cause losses to other vehicles. In order to resist above threats and protect the privacy of vehicle, all shared messages are suggested to be transmitted in ciphertext and the identities of group members need to be authenticated and only vehicles that meet the access control policy can obtain shared data. Consequently, how to design a security and efficient mechanism to realize data sharing under the premise of protecting vehicle and data security is a huge challenge for the continuous and rapid development of VSNs [8].

Blockchain is regarded as a promising technology to support data sharing and access control [9]. Different from traditional technologies, blockchain owns the following advantages[10]. (i) Blockchain eliminates the central server for the maintenance and management of the whole network data by the central server, and improves the flexibility of network organization and data sharing. (ii) Due to the distributed and decentralized characteristics of blockchain, individual tampered data cannot be recognized by the whole network, which ensures the correctness, integrity and security of the data stored in the blockchain. (iii) Blockchain uses cryptography mechanism to effectively control the access of stored data and protects the security data information and the privacy of data owner and data sender. As a result, Blockchain is very suitable for the application scenario of data sharing in VSNs.

Generally, data is maintained by multiple nodes in the blockchain and these nodes are usually the data owners and data users. However, in VSNs, the data owners and data users are usually vehicles. As limited the storage, it is difficult to guarantee that vehicles have enough capacity to store all the data in the communication process. Meanwhile, since fast-moving vehicles need to communicate with surrounding entities by wireless, it is unrealistic for VANETs to ensure the stability and reliability of a large number of data stream transmission in the process of blockchain maintenance. Moreover, due to the fact that once an unauthorized user is compromised, it is difficult to track the user and the records of its usage data. Therefore, in data access control, the authentication of data owners and data users is essential before judging whether these users meet the access policy.

### 1.2. Related Works

Before sharing the data, it is considered essential to authenticate the legitimacy of the vehicle identity. According to [11,12], the true identity of the vehicle should not be exposed to any other entity except trusted authority. Consequently, it becomes critical to design an anonymous authentication scheme. In proposed pseudonym authentication schemes, public key infrastructure (PKI)-certificate based scheme is the most popular authentication scheme [13,14]. PKI scheme requires Central Authorities (CA) issues certificates for vehicles and RSUs to support vehicle-to-infrastructure (V2I) and vehicle-to-vehicle (V2V) authentication within the network. However, according to [15], it is difficult for PKI based on authentication schemes to overcome the following limitations. (1) The adversaries are able to do Dos attacks through invalid signed message; (2) It is difficult to protect the location privacy of vehicles; (3) High computation cost and communication cost. In order to vehicles' location privacy, scheme [16] proposed an anonymous authentication scheme based on social spots (KPSD). KPSD adopts the BonehâĂŞBoyen short signature to achieve the conditional privacy preservation authentication. In social spots, such as the road intersection ,free parking lots near the shopping mall, vehicles are able to generate their pseudonyms and short-life keys independently and implement anonymous

authentication. However, high computational cost leads to low efficiency due to the weak computation capability of vehicles. Zhu etc. adopt group signature to propose a conditional privacy preserving authentication scheme [17]. In the proposed scheme, VANETs are divided into several domains, the group public key of each domain is generated by trusted authority (TA). Vehicles participate in authentication and communication as group members. Scheme [18] uses short group mechanism to protect vehicle privacy. This scheme utilizes cooperative message authentication protocol to achieve distributed key management and alleviate computation overhead. However, due to the indistinguishability of the group signature, once the malicious behavior of the vehicle is found in the group, it is difficult to revoke the vehicle. In order to solve the above problem, schemes [19–22] use identity-based mechanism to provide identity authentication and message verification. These mechanisms eliminate the verification of the certificate, improves the authentication efficiency, and reduces the management overhead of certificate revocation list (CRL). In addition, scheme [21] supports batch authentication and improves the service ratio of RSU. Scheme [22] combines identity based cryptography and group signature mechanism and provides unconditional privacy under the full key exposure attack. However, identity-based signature authentication schemes usually have to face high computational cost. It is a challenge to design an efficient authentication mechanism.

In proposed data sharing schemes in VSNs, scheme [23] proposes a on-demand data dissemination scheme in VANETs, if a vehicle want to obtain data it interested in, it is required to send the request message with beacons. When other vehicles hold the data, these vehicles will dynamically adjust the location of data transmission and send data repeatedly, which are able to effectively protect the location privacy of vehicles. However, the effectiveness of the scheme depends entirely on the density of vehicles in the area. In addition, if there is no incentive mechanism, it is difficult to ensure that the data owner can send the required data to the data user in time. Scheme [24] proposes a access control access scheme based on a decentralized CP-ABE, the proposed scheme support policy hidden and are able to effectively protect the identity of the data owner. However, scheme [24] does not consider the identity authentication of data ownesr and data users, which is not safety to the whole network, because illegal users have a great probability to meet the policy of attribute based encryption scheme. Scheme [25] proposes a verifiable scheme to achieve one-to-many data sharing. The proposed scheme adopts blockchain to achieve access control and guarantee non-repudiation. Meanwhile, policy hiding strategy is designed to hide the privacy of data owner. However, in scheme [25], the blockchain storing data is maintained by vehicles, and it is difficult to determine the method to determine consortium blockchain members. In addition, due to the low storage capacity and computing power of vehicles, it is not practical for vehicles to realize data sharing in time or maintenance through consistency mechanism. Scheme [26] designs a secure vehicle-to cloud service communication mechanism, blockchain is adopted to store reward and punishment records about data sharing. Besides, in order to protect vehicles communication security from malicious vehicles, relevant tracking strategy is also to be proposed. However, the proposed scheme does not mention access policy, the access control of uploaded data depends on CSP without considering the wishes of the data owner.

### 1.3. Contributions

In order to solve above problem, this paper proposes an effective data sharing scheme based on blockchain in VSNs. In the proposed scheme, we first describes the details of anonymous authentication and the establishment of secure communication channels. Then all data is transmitted to the cloud through secure channel. Cloud service provider (CSP) is responsible for managing cloud resources and supports a variety of application services based on cloud resources. Blockchain is adopted to achieve access control and data index. RSUs as the nodes of blockchain save the key words of data and the address where the data is saved in cloud. Only vehicles meeting the access control strategies can obtain the required data from cloud through CSP. In addition, the process of data submission and

data use is saved in blockchain as historical records, and the vehicles that release malicious data and use data maliciously will be tracked in time. The contributions of the proposed scheme are summarized as follows:

1. A secure anonymous authentication protocol is proposed to establish the trust relationship between RSU and vehicle, which realizes the legality verification of communication entity before data sharing.
2. We use blockchain technology and cloud storage to realize the data sharing among vehicles in VSNs, so as to ensure that the data users can obtain the data information they are interested in in time.
3. The proposed scheme supports sensitive hidden information to ensure that data users can not find the sensitive information of the data owners through the obtained data.
4. Security analysis and performance analysis show that our scheme is secure and effective.

### 1.4. Paper Organization

The rest of this paper is structured as follows: Section 2 sketched necessary preliminaries such as VSNs, blockchain, bilinear maps. The details of the scheme are described in Section 3. Sections 4 and 5 discuss the result of the proposed scheme in security and performance respectively. Finally, in Section 6, the conclusions are given.

## 2. Preliminaries

### 2.1. Vehicular Scoial Networks (VSNs)

Vehicular social networks (VSNs) integrate social networks into VANETs and provide a variety of application services for vehicles. Compared with traditional VANETs, VSNs inherit the relevant features of the social networks and provide more humanized service for vehicles [27]. In VANETs, roadside units (RSUs) and vehicles are considered as the main communication entities. As roadside infrastructures, RSUs are deployed on both sides of the road and provide reliable network and communication services for vehicles running within range of signal. Vehicles deployed with on-board units (OBUs) are able to record the running state of vehicles and communicate with nearby vehicles and RSUs. Thus, vehicle-to-infrastructure (V2I) and vehicle-to-vehicle (V2V) are the two main communication forms of VANETs [28]. In V2I, vehicles can communicate with backbone network through RSUs and obtain services or required data from CSP. V2V guarantees that vehicles can obtain the surrounding traffic flow status and other necessary information through the communication with other surrounding vehicles, so as to ensure the safe and smooth driving of the vehicle. In addition, V2V supports the communication between vehicles with similar geographical location and provides more humanized service. In the United States, the standards of V2V and V2I (DSRC/WAVE) are formulated by the Institute of electrical and electronics engineers. Although DSRC/WAVE supports TCP and IP protocols, it is recommended to use the standard called WAVE Short Message Protocol (WSMP), which is more suitable for VANETs network characteristics and communication environment, and ensures the speed of data transmission and processing [3,29]. Social network is considered to be a virtual social relationship network. As social network is integrated into VANETs, VSNs have the capacity to analyze individuals' social relationship through communication information in VANETs, and support related data services to extend drivers' social activities. Therefore, the main service target group of VSN is the people with common interests in given scenarios and time. According to [30], VSNs have supported multiple applications, such as navigation, health-care, safety warming, smart calendar etc.

### 2.2. Blockchain

Blockchain is widely known with the cryptocurrency called bitcoin, which is considered as a new technology to combine decentralization, distributed computation, modern cryptography, and consensus algorithm [31]. As a distributed ledger, blockchain has the following advantages.

1.    Decentralization. Decentralization means there is no need for a third party to centrally manage the system. Due to distributed account and storage, the rights and obligations of any node in system are equal. The data blocks in the system are jointly maintained by the nodes in the whole system.
2.    No tampering. In blockchain, individual tampering cannot be recognized by the whole network, which makes data tampering impossible.
3.    Openness. Blockchain data opens to all nodes except the protected private information. Anyone can query the data stored in blockchain and develop applications.
4.    Auditability. The operation information of nodes is required to store in blockchain and all nodes in the system hold the copy of all data saved. Thus, all logs of users' operations on the blockchain can be queried.
5.    Fault tolerance: Any faults can be corrected by decentralized consensus. If a node fails, blockchain support other nodes to recover all data stored by the fails node.

Figure 1 shows the details of the data structure of bitcoin based blockchain. Each block is divided into block header and block body. Block header stores the hash value of the previous block, root hash, etc. Root hash is the value of merkle root hash, where merkle composes of all transactions stored in blockchain and corresponding hash value. If a transaction on the blockchain is tampered with, the root hash will also be changed, resulting in changes in the content of the whole subsequent block.

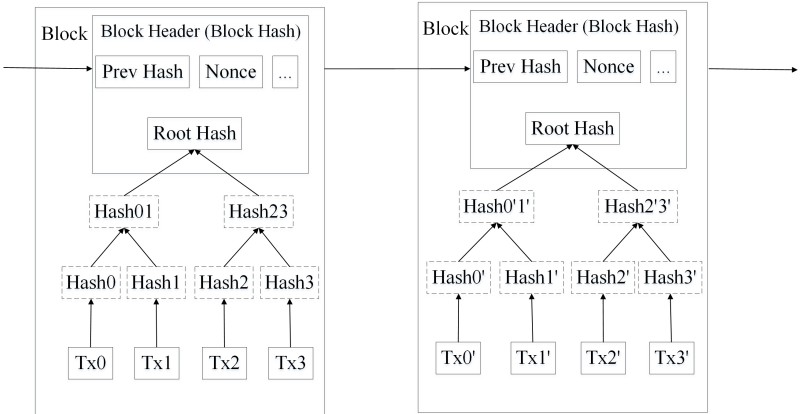

**Figure 1.** Blockchain architecture.

*2.3. Bilinear Mapping*

Support $G_1$ and $G_T$ to be the two groups with the large prime number order $q$, where $G_1$ is additive group and $G_T$ is multiplicative group. A bilinear mapping $e$: $G_1 \times G_1 \to G_T$ satisfies the following properties[32].

1.    Bilinearty: For any $P, Q \in G_1$ and $a, b \in Z_q^*$, $e(aP, bQ) = e(P, Q)^{ab}$.
2.    Non-degeneracy: Existing $P, Q \in G_1$ satisfies $e(P, Q) = 1$.
3.    Computability: For all $P, Q \in G_1$, $e(P, Q)$ can be calculated efficiently.

## 3. The Proposed Scheme

This section gives the details of our scheme, which contains system model, security assumption, security goals, system initialization, initialization registration, V2I authentication, data sharing, and data revocation. The abbreviations which are used in the following protocol are shown in Table 1.

**Table 1.** Symbol and description.

| Symbol | Description |
|---|---|
| $ID_e$ | Entity $e$'s real identity |
| $PS_i$ | The $i$-th pseudonym of entity $e$ |
| $N_i$ | The $i$-th challenge value |
| $Sign_e$ | The signature generated by entity $e$ |
| $C_e$ | The ciphertext encrypted by entity $e$ |
| $K_{e1-e2}$ | The shared key between entity $e1$ and $e2$ |
| $PK_e$ | The public key of entity $e$ |
| $SK_e$ | The private key of entity $e$ |
| $EXP_i$ | The expiration of pseudonym $PS_i$ |
| $TS$ | Current timestamp |
| $A_i$ | The $i$-th attribute |

*3.1. System Model*

As shown in Figure 2, the system model of our scheme consists of four entities, which includes trusted authority (TA), cloud service provider (CSP), RSUs, and vehicles.

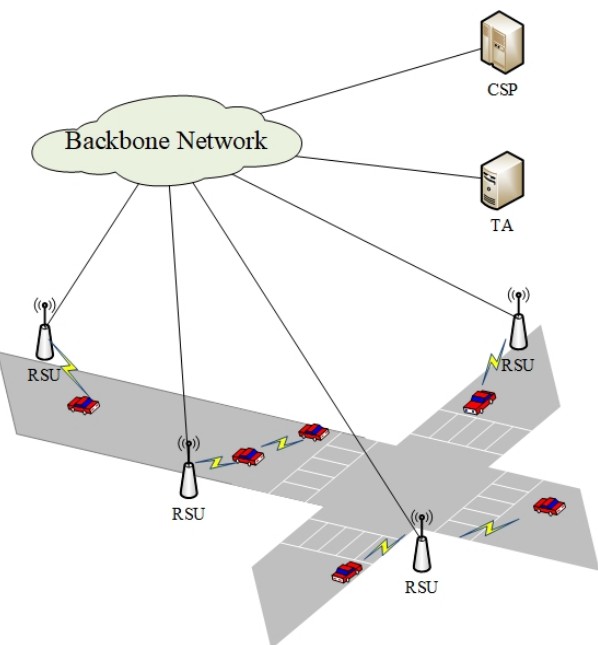

**Figure 2.** System model.

- TA is a third-party trusted authority. All entities in VSNs trust TA. In system initialization, TA is responsible for generating public system parameters, providing registration services for other nodes in VSNs, and supports for establishing trust relationship between vehicles and RSUs.
- CSP is the entity managing the cloud resources and provides a variety of application services based on cloud resources for vehicles in VSNs. In the proposed scheme, CSP provides data sharing services for vehicles to support vehicles to obtain interested data in time.
- RSUs deployed on both roadside have the ability to obtain the surrounding road information by communicating with vehicles, so as to support the vehicles to obtain the necessary information in time. At the same time, RSUs assist vehicles to communicate with CSP to upload or use data. Moreover, all RSUs in VSNs build blockchain network, which stores data owner pseudonym, the key words of shared data, and the address of storing data in CSP. Each RSU shares the data with other RSUs through the consensus mechanism. The vehicle authenticated by RSUs can obtain the address of

the required data stored in CSP through communication with RSUs, and then obtain the data.
- Vehicles follow the WAVE/DSRC standard to communicate with surrounding vehicles and RSUs. In VSNs, vehicles can apply to upload data to cloud through RSU and CSP, and legitimate vehicles can also obtain interested data information from cloud.

### 3.2. Security Assumption

In order to ensure the safety and reliability of data, the following security assumption is made.

**Assumption 1.** *In the proposed effective data sharing scheme based on blockchain in VSNs, we assume that that unauthenticated vehicles are illegal vehicles, which means that these illegal vehicles can not upload data or obtain data they are interested in.*

**Assumption 2.** *In our scheme, we assume that the CSP have access granted to the data, which means CSP may use its access to obtain data and analyze the privacy information of the data owner.*

**Assumption 3.** *RSUs and vehicles are easy to be attacked by adversaries. Before mutual authentication, RSU and vehicle cannot trust the data sent by each other.*

**Assumption 4.** *In the blockchain, RSUs may be compromised by malicious adversaries and become Byzantine nodes. We assume that the number of Byzantine nodes no more than $(n-1)/3$, where n is the number of RSUs in VSNs.*

### 3.3. Security Goal

(1) User Privacy. The true identities of vehicles are hidden from CSP, RSUs, and other vehicles, which means RSU cannot get the true identities of vehicles in authentication and providing services. Meanwhile, data users can not determine the real identity of the data sharer according to the data obtained.
(2) Data Confidentiality. Entities that do not meet the access policy cannot obtain any information related to plaintext through ciphertext.
(3) Accountability and Credential Revocation. All operations on data sharing should be recorded. Once illegal behaviors are found, illegal entities should be revoked in time.
(4) CSP Attacks Resistance. On the basis of guaranteeing data confidentiality, CSP cannot forge or tamper with data.
(5) Minimum Disclosure and Unlinkability. The data users cannot associate with the true identity of the data owner through the acquired data.
(6) Distributed Resolution Authority. Any single authority can not track a vehicle's trajectory or all its behaviors.

### 3.4. System Initialization

In system initialization, TA needs to generate public system parameters and supports to build VSNs security system. The details are shown as follows.

1. Let $G_1$ be an additive group where $|G_1| = p$ for prime $p$, $|G_T|$ be an multiplicative group with the same prime $p$. $P$ is the generator of $G_1$. Meanwhile, An bilinear pairing $e : G_1 \times G_1 \to G_T$ is selected
2. Two hash functions $H_1 : \{0,1\}^* \to G_1$, $H_2 : \{0,1\}^* \times G_T \to Z_q^*$ are defined. TA chooses master key $SK_{TA} \in Z_q^*$.
3. TA compute its public key $PK_{TA} = SK_{TA}P$.
4. For $k$ attributes $\{A_1, A_2, ..., A_k\}$, $K = 3k$ attributes values are defined, where each attribute $A_i$ includes 3 values: $A_i^+$: a vehicle has $A_i$; $A_i^-$: a vehicle does not a proper the attribute $A_i$, and $A_i^*$: $A_i$ does not care.
5. TA chooses $\alpha \in Z_q^*$ and computes $P_i = P^{(\alpha^i)}$, where $i \in \{1, 2, ..., K, K+2, ..., 2K\}$. Then TA selects $\beta \in Z_q^*$ and gets $\omega = g^\beta$.

6. TA broadcasts the parameters $para = \{G_1, G_T, p, e, P, P_1, P_2, ..., P_{K+2}, ..., P_{2K}, PK_{TA}, \omega, H_1, H_2\}$ to all entities in VSNs.

### 3.5. Vehicle Registration Protocol

Vehicles are requested to send their identities to apply for registration. The details are depicted as follows.

1. Vehicle chooses $a \in G_1$, random number $N_1 \in Z_q^*$ and uses $PK_{TA}$ to compute $C_{v-TA} = Enc\_PK_{TA}\{ID_v, N_1\}$ and $aP$. Finally, $C_{v-TA}$ and $aP$ are sent to TA.
2. If receiving the registration message from vehicle, TA first decrypts $C_{v-TA}$ to get $ID_v$, $N_1$. Then, TA generates $n$ pseudonyms $PS_i$ $(0 < i <= n)$ and computes corresponding public key $PK_i = H_1(PS_i || EXP_i)$, private key : $SK_i = SK_{TA} H_1(PS_i || EXP_i)$, where $i \in \{1, 2, ..., n\}$, $EXP_i$ is the expiration of $PS_i$. After that, For the vehicle attribute list $L_v = \{L_v[i], i \in [1, k]\}$, TA picks $k$ random numbers $r = \{r_i \in Z_q^*, i \in [1, k]\}$, and computes $r' = \sum_{i=1}^{k} r_i$. Then, TA generates vehicle attribute based private key $SK_a = \{D, \{D_i\}, \{F_i\}\}$, where $D = \beta^r$, $D_i = P^{\beta(\alpha^{L_v[i]} + r_i)}$. Finally, TA calculates $K_{TA-v} = SK_{TA} aP$ and adopts AES mechanism to encrypt $PS_i$, $SK_i$, $EXP_i$, $SK_a$, $N_1$: $C_{TA-v} = Enc\_K_{TA-v}\{PS_i, SK_i, EXP_i, SK_a, N_1\}$. TA sends $C_{TA-v}$ to vehicle.
3. When obtain the cipthertext from TA, vehicle first computes session key: $K_{v-TA} = aPK_{TA}$, and uses $K_{v-TA}$ to decrypt $C_{TA-v}$ and gets $PS_i$, $SK_i$, $EXP_i$, $SK_a$, $N_1$. vehicle verifies the correctness of $N_1$, if $N_1$ is correct, then vehicle stores $PS_i$, $SK_i$, $EXP_i$, $SK_a$.

### 3.6. RSU Registration Protocol

In this section, RSU register with the TA to obtain its private key. Similar to vehicle registration protocol, RSU first sends its real identity $ID_{RSU}$ to TA. TA computes and returns the private key of $ID_{RSU}$: $SK_{RSU} = SK_{TA} H_1(ID_{RSU})$ to RSU through secure channel. Once receiving $SK_{RSU}$, RSU is able to generate an approved signature and participate in authentication.

### 3.7. V2I Authentication Protocol

When vehicle enters the signal coverage range of RSU, in order to realize the data exchange, vehicles and RSUs are required to use Hess signature mechanism [33] to execute V2I authentication protocol. The details are described as Figure 3.

1. Vehicle chooses $PS_i$, $SK_i$, $EXP_i$, $P_1 \in G_1$, and $r \in Z_q^*$ to generate signature $Sign_v = Sign\_SK_i\{PS_i, EXP_i, TS_1, N_2, r_vP\} = \{h, W\}$, where $h = H_2(PS_i || EXP_i || TS_1 || N_2 || r_vP, e(P_1, P)^r)$, $W = rP_1 + hSK_v$, $TS_1$ is current timestamp, $N_2$ is challenge value and $r_vP$ is the key agreement parameter.
2. Vehicle sends $PS_i$, $EXP_i$, $TS_1$, $N_2$, $r_vP$, and $Sign_v$ to RSU.
3. When receiving the request message from vehicle, RSU first checks the freshness of $TS_1$ and the validity of $EXP_i$. If $TS_1$ is fresh and $EXP_i$ is valid, then, RSU computes $T = e(W, P)e(H_1(PS_i || EXP_i), -PK_{TA})^h$, and check $h == H_2(PS_i || EXP_i || TS_1 || N_2 || r_vP, T)$, if the equation holds, vehicle is considered as a legal vehicle. Finally, RSU signs $ID_{RSU}$, $TS_2$, $N_3$, $r_{RSU}P$ to get $Sign_{RSU} = Sign\_SK_{RSU}\{ID_{RSU}, TS_2, N_3, r_{RSU}P\}$ and generates session key $K_{RSU-v} = r_{RSU} r_v P$. Then RSU adopts AES mechanism to encrypt $N_2$ and gets $C_{RSU-v} = Enc\_K_{RSU-v}\{N_2\}$, where $r_{RSU} \in Z_q^*$.
4. RSU sends $ID_{RSU}$, $TS_2$, $N_3$, $r_{RSU}P$, $Sign_{RSU}$, and $C_{RSU-v}$ to vehicle.
5. Vehicle checks the freshness of $TS_2$ and verifies the legitimacy of $Sign_{RSU}$. If $TS_2$ is fresh, and $Sign_{RSU}$ is legal, RSU is thought to be a legal entity. Then, vehicle generates session key $K_{v-RSU} = r_v r_{RSU} P$ to decrypt $C_{RSU-v}$ and gets $N_2$, if $N_2$ is legal, vehicle believe that a secure channel is established between the vehicle and the RSU. Finally, vehicle adopts AES mechanism to encrypt $N_3$: $C_{v-RSU} = Enc\_K_{v-RSU}\{N_3\}$.
6. Vehicle sends $C_{v-RSU}$ to RSU.
7. RSU decrypts $C_{v-RSU}$ to get $N_3$. If $N_3$ is legal, RSU believes a secure channel is established.

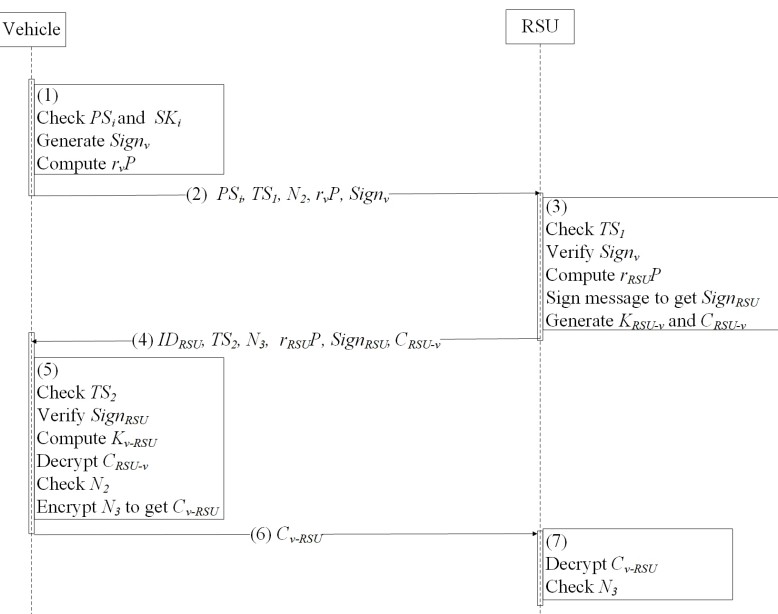

**Figure 3.** Vehicle-to-infrastructure (V2I) authentication protocol.

### 3.8. Data Sharing Protocol

After V2I authentication, vehicle is able to upload or download the data it is interested in. The proposed scheme adopts Zhou's encryption mechanism [34] to achieve the goal of access control. The details are depicted as follows.

As shown in Figure 4, when a vehicle is the data owner, the vehicle are able to upload the data it wants to share to CSP through RSU.

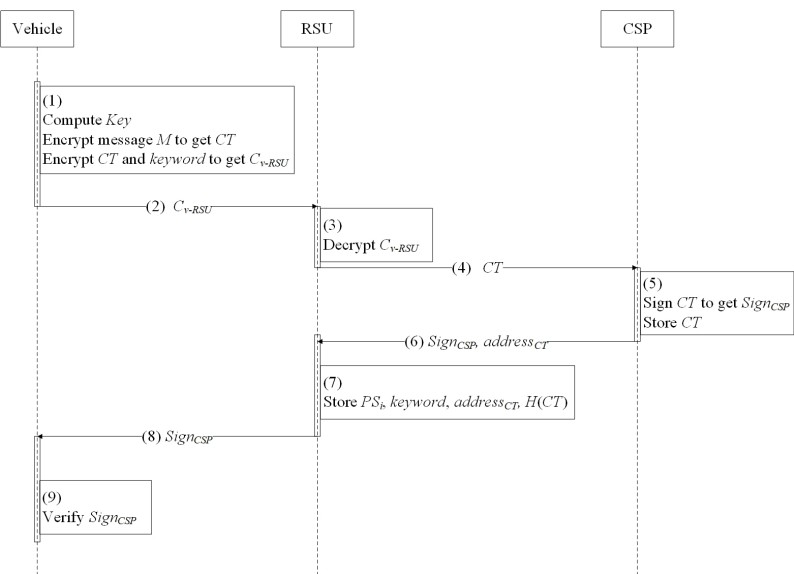

**Figure 4.** Data uploadng protocol.

1. For message $M$ and the policy $W$ with $k$ attributes, vehicle chooses $t \in Z_q^*$ and computes data encryption key $key = e(P_K, P_1)^{kt}$.
2. Vehicle adopts AES mechanism to encrypt $M$ to get $C_v = Enc\_key\{M\}$. Meanwhile, vehicle sets $C_0 = P^t$, $C_1 = (\omega \prod_{j \in W} P_{K+1-j})^t$. Finally vehicle outputs $W' = W \bigcap \{A_i^*\}_{i \in [1,k]}$. The encryptext of $M$ is $CT = \{W', C_v, C_0, C_1\}$.
3. Vehicle encrypts $CT$, the keyword of message $M$ $keyword$ to get $C_{v-RSU} = Enc\_K_{v-RSU}\{CT, keyword\}$, where AES mechanism is used as encryption mechanism.
4. Vehicle sends $C_{v-RSU}$ to RSU.

5.　When RSU receives the ciphertext from vehicle, RSU decrypts $C_{v-RSU}$ to get $CT$, *keyword*, and sends $CT$ to CSP.

6.　CSP stores $CT$ and signs $CT$ and *addr* to gets $Sign_{CSP} = Sign\_SK_{CSP}\{CT, addr\}$, where *addr* is the address of data stored in cloud. Then CSP sends $Sign_{CSP}$ to RSU.

7.　RSU verifies $Sign_{CSP}$ and stores $PS_i$, *keyword*, *addr*, and $H(CT)$ hash value of $CT$ in blockchain. Then RSU sends $Sign_{CSP}$ to vehicle.

8.　Vehicle verifies $Sign_{CSP}$, if $Sign_{CSP}$ is legal, vehicle believes that $CT$ has been stored in cloud.

When a vehicle is the data user, the vehicle can obtain the message it interested in from CSP through RSU. The details are shown in Figure 5.

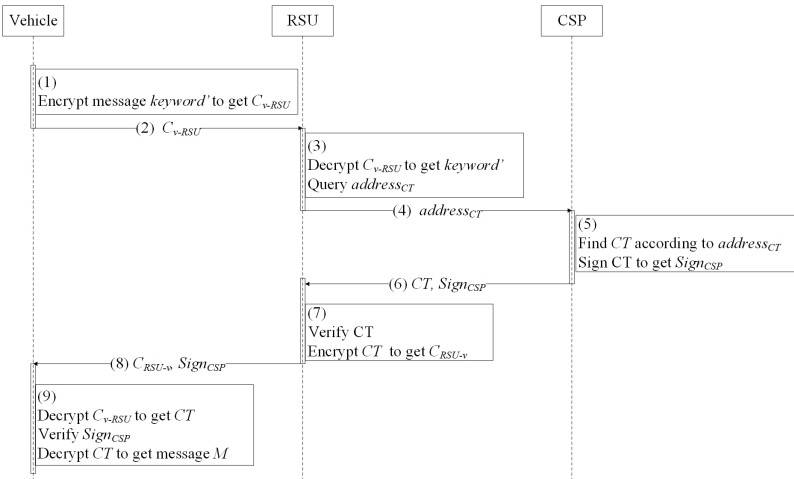

**Figure 5.** Data downloading protocol.

1.　Vehicle adopts AES mechanism to encrypt *keyword'* it interested in to get $C_{v-RSU} = Enc\_K_{v-RSU}\{keyword\}$.

2.　Vehicle sends $C_{v-RSU}$ to RSU.

3.　RSU decrypts $C_{v-RSU}$ and gets *keyword*. Then RSU finds *addr* of the data according to *keyword*.

4.　RSU sends *addr* to CSP via secure tunnel.

5.　CSP queries $CT$ by address and signs $CT$ to get $Sign_{CSP} = Sign\_SK_{CSP}\{CT\}$.

6.　CSP sends $CT$ and $Sign_{CSP}$ back to RSU.

7.　RSU verifies $Sign_{CSP}$ and checks whether the hash value of $CT$ stored in blockchain equals $H(CT)$, if the equation holds, RSU encrypts $CT$ to get $C_{RSU-v}$ and stores the data downloading log into blockchain.

8.　RSU sends $C_{RSU-v}$ and $Sign_{CSP}$ to vehicle.

9.　Vehicle decrypts $C_{RSU-v}$ to get $CT$. Then, vehicle verifies signature $Sign_{CSP}$, if the signature is legal, vehicle constructs local guess of access policy $\tilde{W}$, after that, $\forall i \in [1, k]$, vehicle computes $T_{0,i} = e(P_{\tilde{W}[i]}, C_1)$, $T_{1,i} = e(D[i] \prod_{j \in \tilde{W}, j \neq \tilde{W}[i]} P_{K+1-j+\tilde{W}[i]}, C_0)$. Afterwards, vehicle computes $T_{0,i}/T_{1,i} = e(P, P)^{-t\beta r_i + t\alpha^{K+1}}$. When computing all $k$ terms, vehicle are able to get $key = e(P, P)^{-t\beta(r_1+r_2+..r_k)+kt\alpha^{K+1}} \cdot e(D, C_0)$. Finally, vehicle decrypts $C_v$ to get the message it interested in.

## 4. Security Analysis

This section presents the the security analysis in the following aspects.

(1)　User Privacy. For vehicle identity privacy protection, in mutual authentication, a vehicle uses its pseudonym $PS_i$ and signature $Sign_v$ to prove the legality of its identity in VSNs, which means that no entity other than the TA can determine the true identity of other vehicles. In the process of data downloading, a data user only needs to prove that he/she has the right to obtain the required data, and meanwhile, since data does

not contain any identity information, the data user cannot associate the real identity of the data owner according to the data.

(2) Data Confidentiality. The data is encrypted and stored in the CSP. Any user who does not meet access policies cannot obtain the plaintext. In addition, since the blockchain maintained by RSUs only stores the mapping relationship between keyword and address, RSUs cannot obtain useful data information.

(3) Accountability and Credential Revocation. In a secure network environment, the system can track the data information sent by vehicles in time, and exclude illegal vehicles from the network. The proposed scheme supports illegal vehicles revocation. If a vehicle is comprised, RSU are able to upload its pseudonym $PS_i$, signature $Sign_v$, and operation logs to TA and applies to reveal the true identity of the comprised vehicle. due to the signature and unforgeable logs, the vehicle cannot deny its illegal behaviour. Consequently, when the information of the illegal vehicle is broadcast in VSNs, the illegal vehicle can not communicate with other entities in VSNs.

(4) CSP Attacks Resistance. According to security assumption, CSP has access to grant the data stored, which means CSP may analyze the stored data and try to obtain the privacy of the data owner. In the proposed scheme, the message is encrypted by attribute-based encryption mechanism, CSP can not decrypt the ciphertext depending on its own attributes. Besides, for the issue that CSP may tamper with data, in data uploading protocol, CSP is required to generate $sign_{CSP}$ to prove that the message was stored in the cloud without being tampered with. In data downloading protocol, RSU is able to check whether $H(CT)$ stored in blockchain is equal $H(CT')$, where $CT'$ is the data from CSP. If the verification fails, the data is considered to be tempered.

(5) Minimum Disclosure and Unlinkability. In data sharing scheme, data users can not reveal information other than what the data owner wants to share. In the proposed scheme, the content of data in CSP is completely determined by the data owner. Therefore, any entity cannot obtain the information that the data owner does not want to expose through the data. In the aspect of data association, the association between data and real vehicle information depends on the security of pseudonym changing mechanism.

(6) Distributed Resolution Authority. In a security network environment, any single entity cannot rely on the information collected by itself to track vehicles. For the proposed scheme, in terms of vehicle identity privacy protection, the mapping between the pseudonym and the real vehicle identity is maintained by TA. However, as the vehicle changes its pseudonym frequently during the communication with the surrounding RSUs and other vehicles, TA can not know the vehicle's trajectory alone. Similarly, RSUs only know the pseudonym information and location information of the current communication vehicle. RSUs can not obtain the real identity and long-term trajectory of the vehicle. In terms of data sharing, CSP only provides data uploading and data downloading services for vehicles and cannot accurately know the identity of the data owner. Similarly, RSUs only maintain the list of keyword address and cannot obtain the real content of the data.

## 5. Performance Analysis

This section gives the details of authentication performance of the proposed scheme compared with KPSD [16], LIAP [21], and IMAEP [22] in the computational and communication cost. Moreover, we use Veins simulation framework and Ethereum to test the data uploading and data downloading performance.

### 5.1. Computational Cost

Computational cost is defined the total computation time of RSU and vehicle in mutual authentication. in this section, cheaper operations of point addition operation, one-way hash function are ignored. We focus on expensive operations. $T_{bp}$ refers to the running time of a bilinear pairing operation, $T_{pm}$ indicates the running time of a point multiplication

operation, $T_{pe}$ is the running time of a point exponentiation operation, and $T_{mpt}$ implies the running time of a map-to-point hash function operation.

In order to test the computational cost of the above operations, we make an experiment by choosing the Pairing-Based Cryptography Library. The benchmark includes 2.6 GHz Intel(R) Core(TM) i7-6700HQ CPU, 2GB RAM, Debian 9.4 operating system. The bilinear pairing is $e : G_1 \times G_1 \rightarrow G_T$, where $G_1$ and $G_T$ are additive and multiplicative group respectively. The curve is defined: $y^2 = x^3 + x \bmod p$, where prime number $p = 512$ bits, Solinas prime number $q = 160$ bits. The experiment results are shown in Table 2.

**Table 2.** Pairing and element functions execution time.

| Symbol | Description | Time (ms) |
|---|---|---|
| $T_{bp}$ | Bilinear pairing function | 1.35 |
| $T_{pm}$ | Point multiplication function | 1.77 |
| $T_{pe}$ | Point exponentiation function | 1.74 |
| $T_{mtp}$ | Hash-to-point function | 4.06 |

In KPSD, vehicle picks random number $sk \in Z_q^*$ as temporary private key and computes the public key $pk = g^{sk}$, where $g \in G_1$. Then vehicle selects $\alpha$, $r_\alpha$, $r_x$, $r_\gamma \in Z_q^*$ and calculates $T_U = U_1^\alpha$, $T_V = A_i V_1^\alpha$, $\delta = \alpha sk \bmod q$, $\delta_1 = U_1^{r_\alpha}$, $\delta_2 = T_U^{r_x}/U_1^{r_\delta}$, $\delta_3 = e(T_V, g_2^{r_x})/e(V_1, U_2^{r_\alpha g_2^{r_\delta}})$, $c = H(U_1||V_1||Y_j||T_U||T_V||\delta_1||\delta_2||\delta_3)$, $s_\alpha = r_\alpha + c\alpha \bmod q$, $s_x = r_x + csk \bmod q$, $s_\delta = r_\delta + c\delta \bmod q$. The certificate of vehicle is set $Cert = \{Y_j||T_U||T_V||c||s_\alpha||s_x||s_\delta\}$. Then, vehicle signs message $M$ to get $sign = g_2^{1/x_i + H(M)}$. When receiving message $M$, $sign$, $Y_j$, and $Cert$, RSU computes $\delta' = U_1^{s_\alpha}/T_U^c$, $\delta_2' = T_U^{s_x}/U_1^{s_\delta}$, $\delta_3' = e(T_V, g_2^{s_x} U_2^c)/e(V_1, U_2^{s_\alpha} g_2^{s_\delta})e(g_1, g_2^c)$. If $c == H(U_1||V_1||Y_j||T_U||T_V||\delta_1'||\delta_2'||\delta_3')$, the certificate is considered to be legal. Then RSU checks whether the equation $e(Y_j g_1^{H(M)}, sign) == e(g_1, g_2)$ is hold, if it holds, the sign and $M$ are accepted, otherwise, vehicle's message is rejected.

In LIAP, vehicle first picks $k \in Z_q^*$, and computes its pseudonym $PID = \{PID_1, PID_2\}$, where $PID_1 = kP$, $PID_2 = ID \oplus H(kPK_{CA})$, $ID$ is the real identity of vehicle, $P$, $PK_{CA}$ are public parameters, $H : \{0,1\}^* \rightarrow Z_q^*$. Then vehicle uses local master keys $m_1$, $m_2$ to generate private keys $SK_1 = m_1 PID_1$, $SK_2 = m_2 H(PID_1, PID_2)$. After that, vehicle signs message $M$ to get $\sigma = SK_1 + h(M)SK_2$ and sends $\{PID, M, PK_R$ to RSU, where $h : \{0,1\}^* \rightarrow Z_q^*$, $PK_R = \{PK_R^1, PK_R^2\}$ is the public key of the last RSU that communicated with the vehicle. When receiving the message $\{PID, M, PK_R, \sigma\}$, RSU checks whether the equation is hold: $e(\sigma, P) = e(PID_1, PK_R^1)e(h(M)H(PID_1, PID_2), PK_R^2)$.

In IMAEP, in order to sign message $M$, vehicle selects a set of identities $ID = \{ID_1, ID_2, ..., ID_n\}$, in which vehicle identity is one member of $ID$. Then, vehicle computes the public keys of $PK_{ID} = \{PK_{ID_1}, PK_{ID_2}, ..., PK_{ID_n}\}$, $PK_{ID_i} = H(ID_i) \in G_1$. Afterwards, vehicles selects random numbers $U = \{U_1, U_2, ..., U_n\} \in G_1$, $r_s \in Z_q^*$, $\alpha \in Z_q^*$ and computes $U = r_s PK_{ID} + \alpha P_{pub} - \sum_{i=1,i\neq s}^n (U_i + h_i PK_{ID_i})$, $W = \alpha P$, $h_s = H_0(M||ID||U_s)$, and $V = (r_s + h_s)SK_{ID} + \alpha P_{pub}$, where $P$ is public parameter, $H_0 : \{0,1\}^* \rightarrow Z_q^*$, $SK_{ID}$ is the private key of vehicle. When receiving the message $\sigma = \{U_1, U_2, ...U_n, V, W, ID\}$, RSU first computes $PK_{ID_i} = H(ID_i)$, $h_i = H_0(M||ID||U)$. Afterwards, RSU requests $T_1$, $T_2$ from key generation center, where $T_1 = e(P, W)^{x^2}$, $T_2 = e(P, W)^x$, $x$ is the private key of key generation center. Finally, RSU checks the equation $e(P_{pub}, \sum_{i=1,i\neq s}^n (U_i + h_i PK_{ID_i}))T_1^{-1} = e(P, V)T_2^{-1}$. If the equation holds, $\sigma$ is considered to be legal.

In the proposed scheme, vehicle signs message $M$ to get $sign = \{h, W\}$, where $h = H_2(M, e(P_1, P)^r)$, $W = rP_1 + hSK_v$. When receiving the authentication request, RSU first computes $T = e(W, P)e(H_1(PS_i), P)^r$, and checks $h = H_2(M, T)$. if the above equation holds, the vehicle is considered as a legal node.

The comparison of computational cost is shown in Table 3. In signature generation phrase, the computational cost of KPSD is $3T_{pm} + 9T_{pe} + 2T_{bp} = 23.67$ ms. LIAP

includes 5 point multiplication operations and 1 hash-to-point function operations, the computational cost is $5T_{pm} + T_{mtp} = 12.91$ ms. IMAEP contains $n + 4$ point multiplication operations and 1 hash-to-point function operation, the total of computational cost is $(n + 4)T_{pm} + T_{mtp} = 11.14 + 1.77n$ ms. The proposed scheme contains 1 bilinear map operation and 2 point multiplication operations, the total of computational cost is $2T_{pm} + T_{pe} + T_{bp} = 6.63$ ms. In signature verification phrase, KPSD needs to take $7T_{pm} + 10T_{pe} + 5T_{bp} = 36.54$ ms to verify signature. LIPA needs to compute 3 bilinear map operations, 2 point multiplication operations, and 1 hash-to-point function operation, the total of computational cost is $2T_{pm} + T_{mtp} + 3T_{bp} = 11.65$ ms. IMAEP is requested to calculate 2 bilinear map operations, $n+1$ point multiplication operation, and 1 hash-to-point function operation: $(n + 1)T_{pm} + T_{mtp} + 2T_{bp} = 8.53 + 1.77n$ ms. The proposed scheme contains 2 bilinear map operations, 1 point multiplication operation, and 1 point exponentiation operation, the total of computational cost is $T_{pm} + T_{pe} + 2T_{bp} = 5.21$ ms. Consequently, the proposed proposed scheme is efficient.

**Table 3.** The computational costs result of each schemes.

| Scheme | Signature Computational Cost (ms) | Verification Computational Cost (ms) |
|---|---|---|
| KPSD | $3T_{pm} + 9T_{pe} + 2T_{bp} = 23.67$ | $7T_{pm} + 10T_{pe} + 5T_{bp} = 36.54$ |
| LIAP | $5T_{pm} + T_{mtp} = 12.91$ | $2T_{pm} + T_{mtp} + 3T_{bp} = 11.65$ |
| IMAEP | $(n + 4)T_{pm} + T_{mtp} = 1.77n + 11.14$ | $(n + 1)T_{pm} + T_{mtp} + 2T_{bp} = 8.53 + 1.77n$ |
| our scheme | $2T_{pm} + T_{pe} + T_{bp} = 6.63$ | $T_{pm} + T_{pe} + 2T_{bp} = 5.21$ |

### 5.2. Communication Cost

This section gives the details of communication cost of our scheme compared with KPSD, LIAP, IMAEP. In the bilinear map schemes with respect to 80-bit security level, the size of each element in $G_1$ is 64 bytes $\times$ 2 = 128 bytes, each element in $G_2$ is $2 \times 20 = 40$ bytes. Moreover, the size of $Z_q^*$ and a timestamp are 20 bytes and 4 bytes respectively. Due to the same traffic-related message in all above related schemes, we focus on the size of signature with pseudo-identity. For KPSD, vehicle is required to send $Cert = \{Y_j||T_U||T_V||c||s_\alpha||s_x||s_\gamma\}$, and $sign$, where $Y_j, sign \in G_2$, $T_U, T_V \in G_1$, $c, s_\alpha, s_x, s_\gamma \in Z_q^*$, the communication cost of KPSD is $2 \times 40 + 2 \times 128 + 4 \times 20 = 416$ bytes. In LIAP, vehicle needs to sends $PID = \{PID_1, PID_2\}$, $PK_R = \{PK_R^1, PK_R^2\}$, and $\sigma$ to RSU, where $PID_1$, $PID_2$, $PK_R^1$, $PK_R^2$, $\sigma \in G_1$, as a result, the total communication cost of LIAP is $128 \times 5 = 640$ bytes. In IMAEP, vehicle transmits $\sigma = \{U, V, W, ID\}$, where $U = \{U_1, U_2, ..., U_n\} \in G_1$, $ID = \{ID_1, ID_2, ..., ID_n\} \in Z_q^*$, $V, W \in G_1$. consequently, the communication cost of IMAEP is $148n + 256$ bytes. In the proposed scheme, vehicle sends $PS_i, TS_i, N_i \in Z_q^*$, and $sign_{SK_i} = \{h, W\}$, $h \in Z_q^*$, $W \in G_1$. The communication cost is $3 \times 20 + 20 + 128 = 208$ bytes. The comparison results of the above schemes in communication cost are shown in Table 4.

**Table 4.** The communication cost of schemes.

| Scheme | Message-Signature | Communication Cost (Byte) |
|---|---|---|
| KPSD | $2|G_2| + 2|G_1| + 4|Z_q^*|$ | 416 |
| LIAP | $5|G_1|$ | 640 |
| IMAEP | $(n + 2)|G_1| + n|Z_q^*|$ | 148n+256 |
| Our scheme | $|G_1| + 3|Z_q^*| + |TS|$ | 208 |

### 5.3. Simulation

This section illustrates the experiment result of data uploading and data downloading. We use Veins to run the vehicular network simulation with road traffic simulator SUMO, SUMO is used to generate the movement of vehicles' pattern under a certain trace [35]. and discrete event network simulator OMNET++, Huangpu District of Shanghai, China is selected as the simulation scenario in Veins as shown in Figure 6. In the simulation scenario,

the number of vehicles is 250, and the running route is generated randomly. Vehicles are requested to broadcast the basic safety message every 300 ms. In addition, Etherum is deployed in Debian 9.4 to test the performance of RSU data query. The smart contract is loaded in Ethereum to control the read and write permissions of the data. The simulation parameters are shown in Table 5.

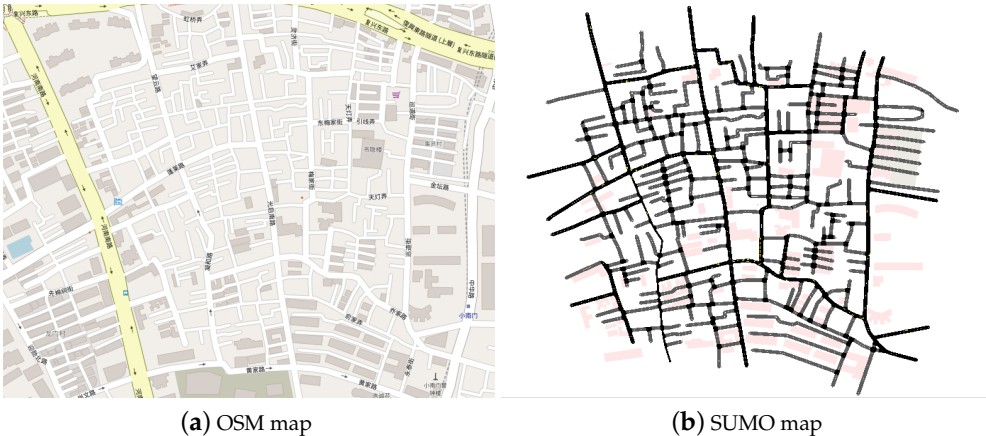

(**a**) OSM map        (**b**) SUMO map

**Figure 6.** (**a**) OSM map of Shanghai (**b**) SUMO NET map of Shanghai.

**Table 5.** Simulation parameters.

| Parameter | Values |
| --- | --- |
| Operating system | Debian 9.4 |
| Traffic generator | SUMO |
| Network simulator | OMNET++ |
| Simulator | veins |
| Simulation area | 1000 m × 1000 m |
| Simulation time | 500 s |
| Number of cars | 250 |
| Data Transmission Rate | 6 Mbps |
| Transmission Power | 20 mW |
| Noise floor | −89 dBm |
| Blockchain | Ethereum |
| The number of attributes | 10 |

Figures 7 and 8 give the result of our scheme in simulation compared with Zhong's scheme [24] and Fan's scheme [25]. Figure 7 shows the total time of data uploading, which includes $T\_Enc_v$: the time for user encryption, $T_{v-RSU}$:the data transmission time between vehicle and RSU, $T_{RSU-CSP}$: the data transmission time between RSU and CSP, and $T_{RSU}$: the time when RSU stores *PS*, *keyword* and *addr* to the blockchain. From Figure 7, we can see that the total data uploading time increases with the increase of vehicles due to the limited communication bandwidth. In Zhong's scheme, when uploading data, vehicle is requested to define an access policy over attributes and encrypt data using encrypt algorithm. As a result, the vehicle needs to execute $2n + 2$ bilinear map operations, $2n + 1$ point multiplication operations, $5n + 4$ point exponentiation operations, and 1 hash-to-point operation, where $n$ is the size of the attributes set. In Fan's scheme, vehicle is required to execute 2 bilinear map operations, 1 point multiplication operations, $2n + 4$ point exponentiation operations, and $n+ 2$ hash-to-point operations to encrypt uploaded data. The proposed scheme requires vehicle, RSU, and CSP to execute 2 bilinear map operations, $n + 4$ point multiplication operations, and 3 point exponentiation operations to upload data. Although the proposed scheme has to meet higher transmission delay due to the participation of RSU and CSP, low computational cost still makes our scheme the most efficient. Figure 8 depicts the average delay of data download, which includes $T\_Dec_v$:

the time for user decryption, $T_{RSU}$: the time when RSU queries *keyword* and *address* from the blockchain, $T_{v-RSU}$, and $T_{RSU-CSP}$. In Zhong's scheme, data user needs to execute $2n$ bilinear map operations, $3n$ point multiplication operations, and $n$ point exponentiation operations to obtain the content of downloaded data. Fan's scheme requires data user to execute $2n + 1$ bilinear map operations, $n + 1$ point multiplication operations, and $n$ point exponentiation operations to get data. In the proposed scheme, data users are required to compute $2n + 6$ bilinear map operations, $n + 3$ point multiplication operations, and 4 point exponentiation operations to obtain the data it is interested in. As a result, the proposed scheme and Fan's scheme are more efficient than Zhong's scheme due to less bilinear maps and point multiplication operations. However, in the proposed scheme, if RSU does not find *keyword'* from blockchain, RSU needs to update its local blockchain through consensus mechanism, which affects efficiency and service ratio of RSU and leads to the proposed scheme owns higher average delay than Fan's scheme. However, in Fan's scheme, since all data is stored in vehicles without the help of RSU, which makes difficult to ensure the data consistency, integrity and security stored in vehicles. Besides, as each vehicle is requested to maintain blockchain and storage data, the computational cost and storage cost of vehicles in Fan's scheme is higher than our scheme even though the average download delay of Fan's scheme is lowest.

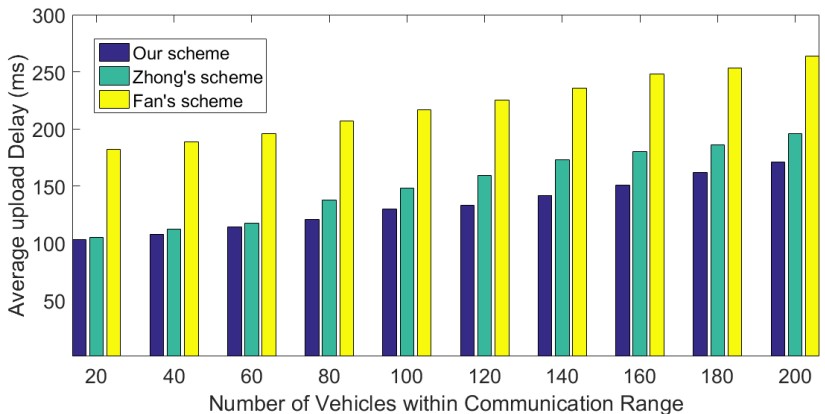

**Figure 7.** Upload data protocol.

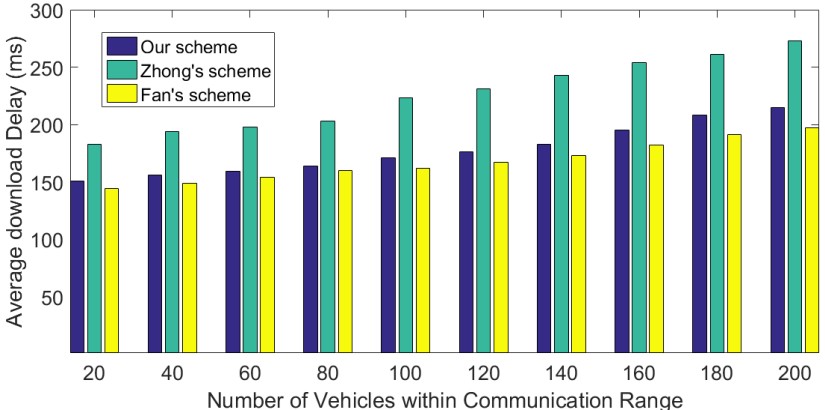

**Figure 8.** Download data protocol.

## 6. Discussion

This paper proposed an effective data sharing scheme based on blockchain in VSNs, which includes anonymous authentication mechanism and data sharing mechanism. In anonymous authentication mechanism, we design a pseudonym generation mechanism and adopt identity based on signature to achieve anonymous authentication between RSU and vehicle. If a vehicle is comprised, TA is able to reveal the real identity depending on the vehicle's pseudonym and corresponding signature. In data sharing mechanism,

RSU is responsible for verifying the legality of vehicles and maintain the key words of data and CSP signature on the data. CSP provides data sharing services for vehicles and supports vehicles to obtain interested data in time. During data uploading and data downloading, CSP is requested to signs the data, which guarantees the data does not be tampered. Consequently, the security of data is also effectively guaranteed. However, the proposed scheme depends on the density of RSUs deployed on both sides of the road, if there is no RSU around the road where the vehicle is travelling, the vehicle cannot upload or download the data of interest. In recent years, although many researchers have proposed vehicle data sharing mechanisms without RSUs, it is difficult to ensure the efficiency of data sharing due to the low computation power and storage capacity of vehicles. In addition, it is still a challenge to ensure the legality and security of data.

## 7. Conclusions

Data sharing is vital for VSNs to provide a variety of humanized services. Meanwhile, the access permissions of shared data should fully consider the wishes of the data owner and the shared data should not expose the privacy of the data owner. We first proposed an anonymous authentication protocol. This mechanism removes the PKI certificate, which not only keeps vehicles low storage, improves the authentication efficiency, but also reduces the management cost of the vehicle. In addition, the pseudonym generation mechanism is adopted to effectively prevent the adversary from obtaining the privacy of the vehicle through tracking. In data sharing protocol, CSP supports the storage and maintenance of data, RSU is responsible for maintaining the blockchain for storing keywords and data addresses, and vehicles are able to upload and download data after mutual authentication. Security analysis shows that the proposed scheme is able to protect the privacy of vehicles and guarantee the confidentiality and integrity of data. Performance analysis proves our scheme is more efficient than traditional schemes.

Due to the limitations of our scheme discussed in Section 6, in the future work, we will focus on researching a more efficient vehicle identity management scheme and propose an efficient data sharing scheme without RSUs.

**Author Contributions:** Methodology, Y.J.; Writing—original draft, Y.J.; Writing—review & editing, X.S. and S.Z. All authors have read and agreed to the published version of the manuscript.

**Funding:** This research received no external funding.

**Acknowledgments:** The work was supported by Liaoning Education Department Science Foundation Project under [Grant Number LJ2020FWL001] and College Students Innovation and Entrepreneurship Training Programs Project under [Grant Number S202010147007].

**Conflicts of Interest:** The authors declare no conflict of interest.

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
