# Peer review of "An Effective Data Sharing Scheme Based on Blockchain in Vehicular Social Networks"

_electronics, doi:10.3390/electronics10020114_

Round 1
Reviewer 1 Report
This paper introduced an anonymous authentication mechanism as the basis for establishing trust relationships before data transmission between entities in VSN. Then a secure and traceable data sharing scheme based on blockchain is described. The proposed scheme is evaluated in terms of security and performance in general setting. More interestingly, the simulation test is also performed. Overall the paper is very well-written and only minor comments I have.
Minor comments:
1. In table 3, many other schemes are compared with the proposed scheme. I just wonder that the functionality is also identical or not to each other.
2. In Figure 7 and 8, the proposed method is slower than Fan's scheme in terms of download data protocol, while upload data protocol is better. Please clearly mention pros and cons of the results.
3. Please mention that what kind of symmetric key cryptography is used or not.
Author Response
Thank you for your suggestion. We have revised our manuscript regarding several issues mentioned above, as follows.
- KPSD, LIAP, and IMAEP in table 3 are anonymous authentication schemes in VANETs, and the aim of these schemes is to achieve anonymous authentication and communication. We have added the details of these schemes in section 1.2 line 84-90, line 101-104.
- We have rewritten the description of Figure 7 and 8 and provided more details of the comparison scheme in section 5.3 line 539-570.
- In vehicle registration protocol and V2I authentication, AES mechanism is adopted to encrypt message after establishing shared key. Besides, in data sharing protocol, data owner adopts AES mechanism to encrypt the uploaded data and gets ciphertext Cv; Data users who meet the access policy can decrypt Cv by AES mechanism to obtain real data. We have added the description of ASE mechanism in section 3.5, line 307, section 3.7 line 334, line 341, and section 3.8 line 354, line 358, line 370.
Thank you for your good evaluation to our manuscripts.
Reviewer 2 Report
The manuscript "An Effective Data Sharing Scheme Based on Blockchain in
Vehicular Social Networks" presents the proposal of an anonymous authentication mechanism as the basis for establishing trust relationships before data transmission between entities in VSNs using secure and traceable data sharing scheme based on blockchain.
Presented topis is very interesting and actual as a research problem, and using blockchain technology for rising security level is welcome.
I recommend several changes for improving the quality of the manuscript:
- give more results information in the abstract
- try to include more current references in the related work section, you can leave out those from 2011 and 2013 unless they are crucial for the paper.
- in 6.2 please start the subsection with text, not a table
- please discuss more your results, especially figure 8. and your results compared with Fan's scheme.
- discuss and highlight more limitations of your solution
- structure of the paper require rewiritening in a more logical order, discussion section should be included, and the overall number of sections should be reduced (when you add Discussion, there would be 8 sections). Try to merge some sections together.
- the conclusion is very brief considering the overall research and work done. Please give more information on your results, and future work (what kind of more efficient vehicle identity management scheme? do you plan to improve the current solution by reducing identified limitations? - check comment No.5. Will you work on a completely new solution?)
Author Response
We would like to sincerely thank you for your advices and constructive comments.
- We have added more details in the abstract to show more results information in line 10-16.
- We have updated references, which includes reference 27-29, 32.
- We have adjusted the position of Table 4 in section 5.3.
- We have rewritten the description of Figures 7 and 8 and provided more details of the comparison schemes in section 5.3 line 539-570.
- We add section 6 discussion, where the limitations of our scheme are described.
- We have combined section 1 and section 2. Besides, we added subtitles in section 1, which makes it easier for readers to understand the paper.
- We have added more information in section 7 conclusion, which includes the limitations of the proposed scheme and future work.
Thank you very much for your suggestions.
Round 2
Reviewer 2 Report
The authors improvements of the manuscript are sufficient and I am satisfied with the authors response. I recommend this paper for accept for publishing in its current form.